# A Comparative Study on the Clinical Efficacy of Simple Transobturator Midurethal Sling and Posterior Pelvic Floor Reconstruction

**DOI:** 10.3390/medicina59010155

**Published:** 2023-01-12

**Authors:** Daoming Tian, Zhenhua Gao, Hang Zhou, Han Lin, Xingqi Wang, Ling Li, Xunguo Yang, Yubin Wen, Quan Zhang, Jihong Shen

**Affiliations:** 1Department of Urology, The First Affiliated Hospital of Kunming Medical University, Kunming 650032, China; 2Yunnan Province Clinical Research Center for Chronic Kidney Disease, Kunming 650032, China

**Keywords:** posterior pelvic floor reconstruction, SUI, surgical effect, TOT

## Abstract

*Background and Objectives:* The purpose of this study was to compare the complications, success rate and satisfaction of pelvic floor reconstruction after transobturator midurethral sling (TOT) and TOT combined with pelvic floor reconstruction in the treatment of female stress urinary incontinence. To explore the pathogenesis of stress urinary incontinence after pelvic floor stress injury and improve the surgical treatment strategy. *Materials and Methods:* From 15 August 2018 to 24 February 2022, patients diagnosed with stress urinary incontinence (SUI) and secondary prolapse of the anterior pelvis were selected to receive surgically. Participants were followed up and evaluated at 2 months, 6 months and 1 year after treatment. According to the patient’s chief complaint, the patient can urinate automatically without incontinence. The number of urinary incontinence and urine leakage was significantly reduced compared with those before operation. Urinary incontinence symptoms did not improve or worsen as ineffective, observing the efficacy and complications. *Results:* We included 191 patients in the TOT group and 151 patients in the pelvic floor reconstruction group after TOT was combined. The operation time and hospital stay in the TOT group were short, but the TOT group needed a second operation to treat recurrent SUI. Perioperative complications were mostly dysuria, and the incidence of postoperative complications in the group of TOT combined with pelvic floor reconstruction was low. The complete success rate and effective rate of pelvic floor reconstruction after TOT in the merger group were significantly higher than those in the TOT group, and the patient satisfaction and complete success rate were also higher. *Conclusions:* TOT combined with posterior pelvic floor reconstruction has a definite short-term effect on patients with SUI and anterior pelvic secondary prolapse. The operation design should pay attention to the support of the posterior wall of the perineum to the bladder neck and the middle and proximal end of the urethra.

## 1. Introduction

Female stress urinary incontinence (SUI) is a widespread chronic disease, and about 200 million people worldwide are affected by urinary incontinence [1]. SUI is the focus and difficult problem in urology and gynecology. Its incidence rate is 18.9%, and the prevalence rate in postmenopausal women is as high as 50% [2]. It can be seen that urinary incontinence has a very wide impact on the lives of women, especially elderly women. Such diseases cause great damage to women’s physical and mental health. Since the 21st century, midstream urethral slings (MUSs) using synthetic materials are the most commonly used surgical techniques for treating SUI [3]. This technology has been highly concerned and deeply studied, and has been widely accepted. This technology reflects the advantages of being minimally invasive, efficient, with fewer complications and having a fast recovery [4]. To a large extent, it is regarded as the “gold standard” for the treatment of female SUI [5,6]. Based on the overall theory of Patros [7], MUSs support the middle part of the urethra by the pubic urethra ligament attached to the pubis, acting as the floor, and squeezing the urethra to maintain urinary control when the intra-abdominal pressure rises. By placing a strip of support (polypropylene mesh belt) in the middle urethra of SUI patients, it can theoretically play the role of reconstructing the floor. At present, a large number of studies in the literature have studied the middle urethral sling operation, especially the Transobturator Midurethal Sling( TOT )operation [8], but there are few studies on SUI with pelvic organ prolapse and posterior pelvic floor repair. In this regard, we specially designed posterior pelvic floor reconstruction to explore the role of the posterior pelvic floor in supporting the urethra. This study compared the treatment of urinary incontinence with POP-Q second-degree patients by pelvic floor reconstruction after Simple TOT and Pelvic floor reconstruction after TOT merger.

## 2. Materials and Methods

### 2.1. Research Object

We retrospectively studied the medical records of 342 patients with SUI combined with anterior pelvic secondary prolapse diagnosed by the Department of Urology of the First Affiliated Hospital of Kunming Medical University from 15 August 2018 to 24 February 2022. The operation used TOT or TOT combined with pelvic floor reconstruction. According to Manonai’s POP-Q grading standard [9], 342 patients were all SUI combined with POP-Q second degree. Exclusion criteria: (1) History of pelvic surgery, dystocia, connective tissue disease, radiotherapy, neuromuscular disease or long-term use of steroids; (2) Physical examination showed that the cervix was prolonged; (3) POP family history and smoking history. Urodynamic examination excluded those with other types of urinary incontinence. Of 342 cases, 285 were women with multiple births as shown in Table 1.

### 2.2. Methods

#### 2.2.1. Preoperative Preparation

Before operation, the patient should be asked about the medical history, physical examination, ultrasound, magnetic resonance, urodynamic examination, etc., and a personalized repair plan should be developed according to the patient’s condition. No fasting is required before operation.

#### 2.2.2. Method of Tension Free Sling Suspension via Obturator

After the anesthesia takes effect, the patient takes the lithotomy position, routinely disinfects the sheets, and retains a 16F catheter. Using 50 mL of 11,000 adrenaline normal saline to make a full water cushion on the front wall of the vagina, and change a 4 cm long straight incision (equivalent to the middle part of the urethra) from 1 cm below the outside of the urethra. Bluntly separate the gap between the urethra and the front wall of the vagina on both sides, and separate the fingertips of both upper branches along the upper edge of the descending branch of the pubis to the rear of the pubis. The left upper branch selects a small round knife 0.5 cm from the outside of the upper edge of the left lower pubic branch to make a 2 mm incision, vertically puncture the above space with a puncture needle, guide the urinary incontinence sling to penetrate from the inside to the outside, check for clear urine color (it is confirmed that there is no bladder damage), complete the implantation of the right urinary incontinence sling arm in the same way, check again for clear urine color (it is confirmed that there is no bladder damage), and place the cystoscope for inspection, No urinary incontinence sling was found to pierce the bladder. The vaginal wall mucosa is intact. Place the tissue scissors into the space between the urinary incontinence sling and the vagina, extract the protective film of the urinary incontinence sling, and lay the urinary incontinence sling flat on the urethra. The tissue scissors can be turned over freely. Fix the sling at four corners with a Belang wire, and fix the urinary incontinence sling at the middle section of the urethra. Trim the excess vaginal mucosa, suture the vaginal submucosal tissue with No. 1 barbed wire, complete the submucosal formation of the anterior vaginal wall, and suture the vagina with absorbable thread to ensure that the vaginal wall does not shrink after forming.

#### 2.2.3. Methods of Posterior Pelvic Floor Reconstruction

Inject water pad on the posterior wall of the canal in the space under the pelvic septum, design a vaginal flap, cut the vaginal mucosa, design the narrowest part of the flap as the pelvic septum, release the flap upward to the cervical plane, vertically suture the rectovaginal fascia, gradually narrow the vaginal canal from the cervical plane to the pelvic septum plane, cross stitch the iliococcygeal muscle, narrow the vaginal genital fissure, and initially form the posterior angle of the vagina. After narrowing and shaping, the vagina only allows two fingers to pass through and sutures the posterior vagina. An absorbable surgical suture was used to continuously suture the vaginal tissue with laceration under the pelvic septum, complete the first layer of perineum shaping, and suture the posterior vagina. Free the skin between the rectum and vagina under the pelvic septum, find the lacerated external anal sphincter, free about lcm to both sides, suture the external anal sphincter continuously with 1 absorbable surgical suture, and further repair and reconstruct the perineum below the vagina. The digital fingertip of the anus is free of blood stain (no rectal injury is confirmed), and the vagina is filled with iodophor gauze to stop bleeding. The sterile dressing covers the wound (Figure 1).

#### 2.2.4. Follow-Up Observation

Record the operation time, bleeding during the operation and urination after operation. Follow up in our department 2 months, 6 months and 1 year after the operation, including whether there are complications after the operation, whether the incontinence symptoms are improved, and whether there is dysuria. Criterion for judging the surgical effect: According to the patient’s chief complaint, the patient can urinate automatically without incontinence. The frequency and symptoms of urinary incontinence were significantly reduced compared with those before the operation. Urinary incontinence is invalid if the symptoms are not improved or aggravated.

## 3. Results

### 3.1. Perioperative Indicators

In the TOT group, 191 patients were hospitalized for 1–4 days, with an average of (2.06 ± 0.74) days. The operation duration was 10–35 min, with an average of (20.5 ± 11.4) min. Intraoperative bleeding was 2–55 mL, with an average of (21.5 ± 12.9) mL. A total of 151 patients in the pelvic floor reconstruction treatment group were hospitalized for 3–7 days, with an average of (3.93 ± 1.05) days. The operation duration was 41–79 min, with an average of (61.04 ± 7.64) min. Intraoperative bleeding was 3–54 mL, with an average of (20.77 ± 10.65) mL. In the group of TOT combined with posterior pelvic repair, posterior pelvic floor reconstruction was added on the basis of TOT alone, and the operation time and intraoperative bleeding were naturally increased, as shown in Table 1.

### 3.2. Surgical Complications

All patients in the two groups successfully completed the surgery. One patient (0.52%, 1/191) in the TOT group punctured and injured the bladder. After using a 2–0 absorbable surgical suture to repair the bladder, a cystoscopy was performed. After operation, the indwelling catheter continued bladder decompression, and the catheter was removed 30 days later. In 2 patients (1.05%, 2/191), the side wall of the vagina was damaged during operation. The side wall of the vagina was repaired with a 2–0 absorbable surgical suture, and the sling was placed again. The follow-up operation effect of the above patients is good without special discomfort, and TOT combined with pelvic floor reconstruction has no operative side injury.

The catheters of the patients were removed 24–48 h after the operation, and all patients were followed up for the first time within 2 months after the operation. One patient in both groups had a urinary tract infection. In the TOT group (0.52%, 1/191), TOT combined with pelvic floor reconstruction (0.66%, 1/151). There was no retropubic hematoma and sling exposure in both groups. The most common postoperative complication was dysuria. Among them, 12 patients in the TOT group had dysuria after operation (6.28%, 12/191). There were only 3 patients (1.99%, 3/151) with dysuria after pelvic floor reconstruction after TOT. We put the catheter in again, and after 5–7 days, the catheter can be pulled out to resume urination. It should be noted that 2 patients in the TOT group had no constipation before operation, and had constipation after operation (1.05%, 2/191), while those in the treatment group combined with pelvic floor reconstruction did not, as shown in Table 2.

### 3.3. Clinical Efficacy

In total, 342 patients were followed up 6 months and 1 year after operation. The patients were asked questions according to the following four criteria: without other treatment after surgery, the SUI symptoms have completely disappeared, and the patient can control urine autonomously. The improvement was that SUI symptoms were less than those before operation, a small amount of urine leaked out during strenuous activities, and there was no urine leakage during daily activities. Ineffective means that SUI symptoms are not improved or further aggravated compared with those before operation [9], and urine leakage is still unable to be controlled. Questions were asked about the degree of satisfaction with the operation, which is divided into the following five levels: very satisfied, satisfied, general satisfied, dissatisfied and very dissatisfied. Very satisfied and satisfied correspond to cure, general satisfaction corresponds to improvement and dissatisfaction and very dissatisfied correspond to invalidity. A complete success rate means that the patient is effective after operation, without intraoperative complications and reoperation. According to statistics, the cure rate of TOT was 86.39% (165/191), the effective rate was 90.58% (173/191) and the complete satisfaction rate was 80.10% (153/191). The cure rate of pelvic floor reconstruction after TOT was 90.73% (137/151), the effective rate was 94.79% (143/151) and the complete satisfaction rate was 92.05% (139/151), as shown in Table 3.

## 4. Discussion

### 4.1. Comparison of Therapeutic Effects of TOT and TOT Combined with Pelvic Floor Reconstruction

Relevant research shows that TOT operation time is only 1/2 of TVT operation time, but the surgical cure rate of both is similar [10], so more and more operators choose TOT to treat stress urinary incontinence. However, the number of patients with simple urinary incontinence without pelvic organ prolapse in clinical observation is relatively small, and the effect of only using a middle sling to treat SUI is not significant. In order to improve the effect of pelvic floor reconstruction, reduce the recurrence rate, restore the biomechanical axis, and complete the physiological mechanical reconstruction, we designed this operation. In our study, we judged that in 342 patients with SUI and POP grade II who were treated surgically, the criteria of efficacy were as follows: without any other treatment after surgery, SUI symptoms had completely disappeared, urination could be controlled automatically, and SUI did not recur after surgery. The improvement is that SUI symptoms are less than those before surgery, a small amount of urine leaks out during strenuous activities, no urine leakage symptoms during daily activities and no recurrence of SUI after surgery. Ineffective means that SUI symptoms are not improved or further aggravated compared with those before operation, and urine leakage is still unable to be controlled [11]. In the simple TOT group, the cure rate was 165/191 (86.39%) and the total effective rate was 173/191 (90.58%), which is similar to previous reports at home and abroad [12]. The cure rate of the treatment group of pelvic floor reconstruction after TOT was 137/151 (90.73%), and the total effective rate was 143/151 (94.79%). Although the difference between the two groups was small, the curative effect of the treatment group of pelvic floor reconstruction after TOT was better than that of the simple TOT group. Compared with other TOT operations reported in the literature, our surgery has a higher cure rate [13]. In the meta-analysis and evaluation of the short-term efficacy and adverse reactions of Ajust single incision micro urethral sling operation (Ajust operation) compared with standard middle urethral sling operation (TVT-O/TOT) by Wei et al., [14] 233 cases in TVT-O/TOT group were reported. Of these, 190 cases were cured, with a cure rate of 81.59%. In addition, we put forward the concept of a complete success rate. The complete success rate means that there is no symptom during the operation, it is effective after the operation, and there is no reoperation. Complete success rates of 153/191 (80.10%) in the pure TOT group and 139/151 (92.05%) in the treatment group of pelvic floor reconstruction after the TOT combination were compared. The difference between the two groups was significant, which further reflected that pelvic floor repair after SUI combined with POP II was the best choice.

Why are there significant differences in complete success rates between the two groups? The key to explaining this phenomenon is surgical design. The treatment group with pelvic floor reconstruction after TOT has a high effective rate and cure rate. We strengthened the perineum through posterior pelvic floor reconstruction. The middle part of the urethra to the external opening of the urethra is supported by the perineum. Athanasopoulos pointed out in his article that the perineum is short if its length is less than 3 cm. The short perineum is related to poor anatomical support of pelvic viscera [15]. At present, it is mainly believed that the perineum is the attachment point of pelvic floor muscles. The perineum is the final line of defense for the distal end of urogenital hiatus to prevent pelvic floor dysfunction diseases. With the increase of its area, gravity will cause bladder bulge, rectum bulge, cervix or uterus prolapse to fill the enlarged space, and the tension of fascia and muscle in the pelvic cavity will increase to maintain the normal position of pelvic organs. In some patients with urinary incontinence, the perineal body was torn and the vagina collapsed due to birth injury, resulting in the widening of the vaginal diameter. Therefore, both the cure rate and the effective rate in the treatment group of pelvic floor reconstruction after TOT were higher than those in the simple TOT group.

### 4.2. Comparison of Complications of TOT and TOT Combined with Pelvic Floor Reconstruction

The treatment group of TOT combined with pelvic floor reconstruction had low complications. Perioperative complications were mostly dysuria. The incidence of dysuria in the TOT group was 2/191 (6.28%), while the incidence of dysuria combined with pelvic floor reconstruction was 3/151 (1.99%). According to 62 randomized studies and a meta-analysis of 7101 women, the short-term cure rate in the TOT group was 73%~82%, and 4%~8% of women receiving obturator sling surgery would have dysuria [16]. The reason why simple TOT surgery often causes difficulty in urination is the tightness adjustment of the sling. The tension-free middle urethral sling is placed too loosely, and the surgical effect is poor. If the sling is placed too tightly, urination will be difficult. Therefore, it is always emphasized that there is no tension, which tests the level of the operator. If the sling is placed too tightly, the front wall of the vagina will be deformed and broken. Therefore, patients with dysuria after surgery often complain that they can urinate smoothly only when they are in a squatting position as shown in Figure 2A. The reason why the patients in the pelvic floor reconstruction group found that the difficulty in urination was less after TOT was combined was that the vaginal levator ani tear supported the bladder neck. Through the reconstruction of the posterior pelvic floor, the vaginal levator ani tear was repaired, so that its resultant force was directed to the bladder neck, and the originally broken urethra naturally recovered to normal physiological morphology after receiving the resultant force as shown in Figure 2B.

Our research suggests that TOT combined with posterior pelvic floor reconstruction can enhance the support of the posterior vaginal wall and improve the urethral closure pressure, thus achieving a good surgical effect. However, due to the limitations of the follow-up conditions in this study, a considerable number of patient follow-ups were completed by telephone, and the criteria for judging the postoperative efficacy were relatively simple. The patient’s satisfaction with the surgical effect may also serve as the patient’s subjective judgment of the surgical effect. In addition, the efficacy of the two groups of patients may be affected by time, so long-term follow-up of patients after surgery is essential.

### 4.3. Mechanism of Female Continence and Role of Posterior Pelvis

The mechanism of female continence is an important anatomical and physiological function. Because its regulatory mechanism is complex and involves pelvic floor biomechanics, there is still a great controversy. Moreover, there are still blind spots in the understanding of the pathogenesis of SUI and POP and the choice of individualized treatment. In the past, it was believed that stress urinary incontinence was caused by lax closure of the bladder neck due to pelvic septum relaxation, and smaller pressure difference due to the shortening of functional urethra [17]. After the establishment of the hammock theory in the 1990s, Petros’ overall theory believed that the pubo urethral ligament was lax. It enables the posterior muscle force to transfer the urethra from the “C” (closed) state to the “O” (open) state. The resistance of the urethra is greatly reduced, and the middle urethra cannot be supported, resulting in urine leakage We found that urinary incontinence belongs to pelvic floor stress disease in the final analysis [7]. Due to various high-risk factors such as vaginal delivery and age, the pelvic floor supporting structure is not fully functional. With the increase of the internal abdominal pressure, the internal abdominal pressure first acts on the top wall of the bladder and transmits downward to the pubic symphysis. Because the pubic symphysis bone structure is not easy to produce elastic deformation, the resultant direction will be transmitted to the posterior corner of the bladder, causing the bottom of the bladder to go down, the posterior corner to disappear, the internal orifice of the urethra to change like a funnel, the functional urethra to become shorter, and finally urine leakage. The vagina passes through the levator ani fissure (urogenital diaphragm) to form an included angle, which is the narrowest part of the vagina and acts as a supporting point for the posterior wall of the bladder. In addition, the posterior pelvis plays an important role in supporting the middle segment of the bladder neck urethra. Due to the collapse of the posterior wall of the bladder, the anterior vaginal wall prolapses. The prolapsed anterior vaginal wall and the urethra will collapse backward to the anterior and posterior vaginal walls through the paranasal space. The damaged perineal body will find it difficult to generate the resultant force of resistance, which will aggravate the occurrence of stress urinary incontinence. The surgical design can strengthen the perineal body according to this, increase its supporting effect, and make the resultant force act indirectly on the bladder neck.

## 5. Conclusions

Although there were differences in hospitalization time, operation time and intraoperative bleeding between the TOT group and TOT combined with the pelvic floor reconstruction group. However, in the treatment group of TOT combined with pelvic floor reconstruction, the effective rate, cure rate, complete success rate and complications were high. The operation was effective. Surgical design should pay attention to the support of the back wall of the perineum and vagina to the bladder neck and the middle and near end of the urethra.

## Figures and Tables

**Figure 1 medicina-59-00155-f001:**
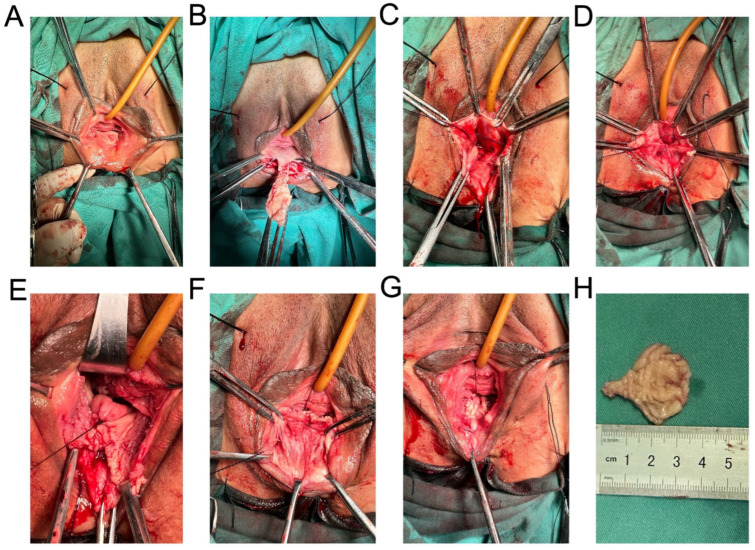
Methods of posterior pelvic floor reconstruction. (**A**) Injection of posterior wall water pad into the space under the pelvic septum. (**B**) Design of vaginal flap. (**C**) Suture the fascia beside rectum and vagina. (**D**) Narrow vaginal genital hiatus. (**E**) Suture the posterior vagina. (**F**,**G**) Suture the external anal sphincter and reconstruct the perineum. (**H**) Vaginal flap.

**Figure 2 medicina-59-00155-f002:**
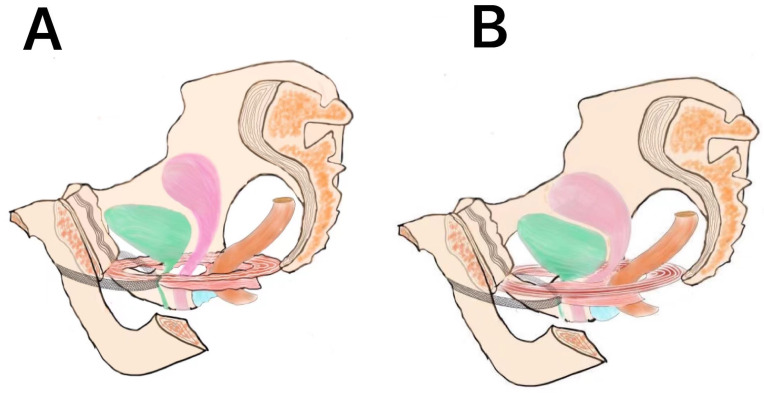
Schematic diagram of operation. (**A**) Dysuria caused by simple tot operation. (**B**) Supporting urethra in the combined pelvic floor repair group.

**Table 1 medicina-59-00155-t001:** Characteristics of the Study Patients.

Characteristics	TOT	TOT Combined with Pelvic Floor Reconstruction	*p* Value
Age (mean ± SD, y)	52.92 ± 11.52	50.43 ± 10.24	<0.05
BMI (mean ± SD, kg/m^2^)	23.55 ± 2.93	23.92 ± 3.04	>0.05
Number of pregnancies (mean ± SD, times)	3.11 ± 1.40	3.47 ± 1.47	<0.05
Hospital stay (mean ± SD, d)	2.06 ± 0.74	3.93 ± 1.05	<0.05
Operation time (mean ± SD, min)	20.5 ± 11.4	61.04 ± 7.64	<0.05
Blood loss (mean ± SD, mL)	21.5 ± 12.9	20.77 ± 10.65	>0.05

Abbreviations: SD, standard deviation; BMI, body mass index; y, year; d, day; min, minute; mL, milliliter.

**Table 2 medicina-59-00155-t002:** Surgical Complications.

Characteristics	TOT	TOT Combined with Pelvic Floor Reconstruction	*p* Value
Bladder injury	1/191 (0.52%)	0/151 (0%)	>0.05
Injury of vaginal wall	2/191 (1.05%)	0/151 (0%)	>0.05
Retropubic hematoma	0/191 (0%)	0/151 (0%)	>0.05
Incision infection	0/191 (0%)	0/151 (0%)	>0.05
Urinary tract infection	1/191 (0.52%)	1/151 (0.66%)	>0.05
Dysuria	12/191 (6.28%)	3/151 (0.66%)	>0.05
Leg pain	2/191 (1.05%)	0/151 (0%)	>0.05
Postoperative constipation	2/191 (1.05%)	0/151 (0%)	>0.05
Exposed sling	0/191 (0%)	0/151 (0%)	>0.05

**Table 3 medicina-59-00155-t003:** Clinical Efficacy.

Characteristics	TOT	TOT Combined with Pelvic Floor Reconstruction	*p* Value
Cure rate	165/191 (86.39%)	137/151 (90.73%)	>0.05
Effective rate	173/191 (90.58%)	143/151 (94.79%)	>0.05
Reoperation rate	4/191 (2.09%)	3/151 (1.99%)	>0.05
Patient satisfaction rate	187/191 (97.90%)	148/151 (98.01%)	>0.05
Complete success rate	153/191 (80.10%)	139/151 (92.05%)	0.002

## Data Availability

Not applicable.

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
