# Peer review of "A Comparative Study on the Clinical Efficacy of Simple Transobturator Midurethal Sling and Posterior Pelvic Floor Reconstruction"

_medicina, 2023, doi:10.3390/medicina59010155_

Round 1

Reviewer 1 Report

The overall manuscript presentation is good. The study is interesting.

However, some corrections are requitted.

Title: Please avoid abbreviations in the title. "TOT"? what if a potential reader is not familiar with the abbreviation?

In the abstract, all abbreviations should be explained at the first appearance. Again "TOT".

Methods - Table 1. Characteristics of the study patients should be moved to the Results section and presented in the subheading 3.1 Study subjects description. 

The results part should include the study subjects' socio-demographic data.

The discussion part is too long. The study's strengths and limitations should be outlined.

Please make sure the discussion part is covering the following required items:

1 Rationale of the study (why it was done)

1.1 Main findings of the study

1.2 What makes our study unique

1.3 What it adds to what we already know

2.Study subjects - Subject of the discussion

2.1 Comparison of your results with previous studies on the same topic, agreement, and disagreement with the studies compared

3. Sum up of the study, study strengths and limitations

4. Clinical implication

Reviewer 2 Report

Dear author's

I was pleased to review your article submitted to Medicina. The subject in not new but the article is well written.

I have the following comments:

1. Please explain the novelty of your study.

2. Knowing this results how you propose to improve patients care?

3. In the section Discussion you it is mandatory to compare your results with the existing data in the literature.

4. Minor English edits.

Round 2

Reviewer 2 Report

Dear author's

Thank you for your response. Your study is well designed and recommend a new approach the combination of TOT with pelvic floor reconstruction. Other studies should confirm the benefit of the combination techniques.